# Checkpoint Inhibitors and Induction of Celiac Disease-like Condition

**DOI:** 10.3390/biomedicines10030609

**Published:** 2022-03-04

**Authors:** Aaron Lerner, Carina Benzvi

**Affiliations:** 1Chaim Sheba Medical Center, The Zabludowicz Research Center for Autoimmune Diseases, Research Department, Ramat Gan 52621, Israel; carina.ben.zvi@gmail.com; 2Research Department, Ariel University, Ariel 40700, Israel

**Keywords:** celiac disease, immune checkpoint inhibitor, Nivolumab, PD-1, Ipilimumab, CLTA-4, gut toxicity, immune toxicity

## Abstract

Immune checkpoint inhibitors herald a new era in oncological therapy-resistant cancer, thus bringing hope for better outcomes and quality of life for patients. However, as with other medications, they are not without serious side effects over time. Despite this, their advantages outweigh their disadvantages. Understanding the adverse effects will help therapists locate, apprehend, treat, and perhaps diminish them. The major ones are termed immune-related adverse events (irAEs), representing their auto-immunogenic capacity. This narrative review concentrates on the immune checkpoint inhibitors induced celiac disease (CD), highlighting the importance of the costimulatory inhibitors in CD evolvement and suggesting several mechanisms for CD induction. Unraveling those cross-talks and pathways might reveal some new therapeutic strategies.

## 1. Introduction

Immune checkpoint inhibitors (ICPis) are a relatively new oncogenic immunotherapy that is based on our understanding of immune cells’ regulatory pathways. Upon stimulation, these key regulators can suppress the immune system and protect the host from unwanted or overreacted responses. This mechanism can be exploited by cancer cells to escape the system that is designated to eliminate them. Indeed, several invasive cancers defend themselves by stimulating checkpoint targets. The new line of ICPis block these immunosuppressive points and restore the immune functionality against the cancerous cells [1,2,3,4]. The currently approved ICPi target the receptors, cytotoxic T-lymphocyte-associated protein 4 (CTLA-4), programmed cell death protein 1 (PD-1), and programmed death-ligand 1 (PD-L1). These represent the basic scientific productive discoveries of James P. Allison and Tasuku Honjo, who won the Nobel Prize in 2018 [5]. The present review will be limited to the adverse effects of those inhibitors, mainly on the auto-immunogenic aspects, zooming on ICPi drug-induced CD. Following are some sub-paragraphs expanding on CD in general, ICPi and autoimmune diseases (ADs) relationships, a summary on environmental and drug induced CD and ICPis mode of action.

### 1.1. Celiac Disease in a Nutshell

Celiac disease or gluten-induced enteropathy is a life-long AD that affects the duodenum and the proximal small bowel, where patients lose their tolerance to gluten [6]. Gluten is the main wheat protein, which is also present in rye, barley, and contaminated oat. In the past, the classic symptoms were gastrointestinal, however, in recent decades, a shift to extraintestinal manifestations is taken place [7], and even obesity and acute presentations were described. There is a genetic susceptibility and various non-gluten environmental factors were identified. The diagnostic criteria were modified for symptomatic children by ESPGHAN [8], but the adult guidelines include intestinal biopsies for definitive diagnosis. As with many ADs, the incidence of CD has been continuously rising in recent decades [9]. There is a plethora of serological markers [10], where the most clinically used is the IgA anti-tissue transglutaminase which is the corresponding IgA against the auto-antigen of CD. It is a multifaced condition that, with clinical unawareness, can be easily undetected.

### 1.2. Potential Risk Modifiers of Celiac Disease

Similar to other ADs, CD has a genetic background and is influenced by several environmental risk modifiers. It is well established and documented that gluten, by far, is the ultimate inducer of CD and some other gluten-dependent conditions. For many years, the timing, the quantity of gluten introduction, and breastfeeding during early infancy were suspected of modifying CD behavior [11]. Several more recent studies failed to confirm those claimed associations [12,13,14,15]. The same is true for cesarean section, as reported by a Swedish and a Canadian group [16,17]. Only a small increase in CD appearance was noted in mothers who underwent an elective section [16]. The results may lead to the importance of the cervical microbiome in mediating CD during the perinatal period. Smoking is a modifier of many chronic metabolic, cancer, and ADs. As concerns CD, the data are inconclusive. Some found that smoking is protective [18], and others found it was a weak contributor [19].

Early life exposure to antibiotics was found to increase the odds of CD development [20,21]. Many associations were reported to relate infections to CD, but no causative relationship was substantiated, though mechanisms and pathways were suggested [22,23]. Celiac disease was associated with a plethora of viruses. Adenovirus 12 E1A, Hepatitis C, rotavirus, and, most recently, reovirus were reported. Intriguingly, the Rotavirus vaccine might be a protective factor [24] but merits further international investigation. Even SARS-CoV-2 was implicated, most recently, in the surge of CD incidence.

No less puzzling is the relation between CD and bacterial infections. Early life and adulthood intestinal infections are linked to CD, but the lack of data makes the interpretation questionable [11,25]. The range of gastrointestinal infections in CD is quite broad and was summarized, highlighting the pros and cons. Interestingly, the vaccination-CD relationship is part of the puzzle of vaccine-induced autoimmunity in the frame of the ASIA syndrome [26,27]. Helicobacter pylori might play a protective role in CD [28], and Candida albicans was recently suggested as a trigger for CD [29].

An adverse influence of pregnancy is known to occur in women with overt or silent CD [30], upon abdominal surgery [31], and emotional stress [32]. It should be stressed that those relations are associative, and no definitive causation was detected. More studies are necessary to disclose the underlying mechanisms in order to develop potential preventive strategies.

Finally, drugs were reported to be linked to CD evolvement. The use of antibiotics in early life was found to enhance the risk of CD later in life [21,33]. Proton pump inhibitors were reported to increase the risk for CD development [34,35]. The authors suggested that the luminal enhancement of gluten immunopathology and the alterations in gut microbiota might drive the phenomena. There is always the possibility that the drugs were given for symptoms caused by undetected CD rather than initiating CD development [11].

### 1.3. Immune Checkpoints

Naïve T cells are activated by antigen-presenting cells (APCs) in secondary lymphoid organs. It requires several signals, including the engagement of T cell receptor (TCR) with an antigen peptide, presented on major histocompatibility complex (MHC); the binding of CD28, expressed on T cells, to B7, expressed on APC; and the release of co-stimulation signals and cytokines from APCs, affecting T cells differentiation and proliferation into various T cells’ subtypes [36].

Immune checkpoints, CTLA-4 and PD-1, are part of an array of molecules that tightly coordinate immune balance and surveillance, regulating an effective immune response against hostile agents while maintaining tolerance against self-antigens. The checkpoints CTLA-4 and PD-1 are involved in controlling autoreactive T cells and preserving peripheral tolerance. They serve as co-inhibitors to fine-tune the immune response and prevent overreaction of the immune system [37,38].

The critical role of CTLA-4 was demonstrated on knockout mice that develop massive lymphoproliferation with severe and fatal multiorgan failure [39,40]. With respect to PD-1, its deficiency in mice leads to a milder disease, resulting in T cells hyperproliferation and autoimmunity that depends on the mice’s genetic nature. In humans, CTLA4 and PD-1 polymorphisms were associated with various ADs, including Grave’s disease, autoimmune hypothyroidism, systemic lupus erythematosus, mild glomerulonephritis, cardiomyopathy, ankylosing spondylitis, rheumatoid arthritis (RA), type 1 diabetes, and CD [41,42,43,44,45,46].

Upon T cells’ activation, CTLA-4 receptors gradually emerge on their membrane. These molecules compete against CD28 with higher affinity to the B7 family of ligands, leading to decreased availability of CD28 co-stimulators [47]. As a result, CTLA-4 receptors bind to most of B7 ligands and pass inhibitory signals into the T cell [48]. Furthermore, FOXP3 regulatory CD4 T cells inherently express CTLA-4; upon interaction with APCs, they release immunosuppressive cytokines, like IL-10 and TGF-β, and they restrain effector T cells co-stimulation [49,50]. The transmembrane protein, PD-1, is found on various immune cells, including B cells, NK, macrophages and dendritic cells. Yet, its functionality in cancer immunotherapy has been explored mainly on T cells. This molecule regulates T cell activation through interaction with programmed death-ligand 1 or 2 (PD-L1 or PD-L2) [51]. Similar to CTLA-4, T cells begin to express PD-1 following their activation. However, while CTLA-4 functions primarily in lymph nodes, PD-1 functionality is chiefly in peripheral tissues, where effector T cells operate without the need for additional stimulation. When encountering acute and chronic infection, particularly during cancer, PD-1 expression increases [52,53]. An engagement with its ligands, PD-L1 or PD-L2, issues downstream signaling, resulting in the reduction in transcription factors such as TBET and GATA3. As a result, effector T cell functionalities are suppressed and proliferation is attenuated through metabolic restrictions, leading to apoptosis, anergy, and exhaustion [37,54,55,56].

### 1.4. Immune Checkpoint Inhibitors in Cancer Therapy

The employment of immune checkpoint pathways by cancer cells serves them to effectively evade immune surveillance [57]. They express inhibitory ligands, PD-L1 and B7, which engage with PD-1 and CTLA-4 on antitumor effector T cells. These interactions induce a cascade of intracellular events, resulting in down-regulation of T cells’ response and ultimately culminating in tumor immune tolerance [58,59].

To overcome this, a therapeutic strategy was developed to arrest these immunosuppressive pathways. The implementation of ICPi was found to be a paradigm shift in cancer treatment. It is based on the principle of blocking negative co-stimulation with monoclonal antibodies, like anti-CTLA4 and anti-PD1/PDL1 (Figure 1). In doing so, the prevention of CTLA-4 or PD-1 from binding to their respective ligands unleashes effector T cells’ responses at the tumor level, reversing their exhaustion, reactivating adaptive and innate functionality and increasing tumor cells elimination [58,60,61]. Nevertheless, the success of these treatments requires the presence of exhausted tumor-specific effector T cells and a tumor microenvironment that highly expresses immunosuppressive signals [51].

Ipilimumab, anti-CTLA-4, was the first ICPi to display high anti-tumor activity in patients with metastatic melanoma, and it was the first FDA approved by 2011 [62]. Following this, Nivolumab and Pembrolizumab, that target PD-1, were found to have fewer toxicities, durable tumor remission, and improved overall survival in the treatment of advanced melanoma [63,64,65]. They were FDA approved for the treatment of additional cancers, including non-Small Cell Lung Cancer (NSCLC), metastatic head and neck cancer, renal cell carcinoma, and Hodgkin lymphoma (HL). The drugs, Atezolizumab, Avelumab, and Durvalumab, target PD-L1, were all FDA approved for urothelial carcinoma and as second-line therapy for NSCLC [66,67]. Since then, checkpoint inhibitors have been approved to treat a broader range of metastatic diseases, and other inhibitory pathways are being explored as potential targets for cancer treatment [68].

### 1.5. Immune-Related Adverse Events with Checkpoint Inhibitors

Despite the unprecedented success of ICPis in treating multiple types of advanced malignant cancers, unbalancing immunoregulatory control can lead to unrestrained self-reactive T cells and an increased risk of developing autoimmune and autoinflammatory toxicities. In the majority of cases, these manifestations, which are referred to as immune-related adverse events (irAEs), outbreak when treated with anti-CTLA-4 and anti-PD-1/PD-L1 antibodies [69].

The severity of irAEs is dose-dependent and patients can potentially develop a spectrum of side effects that vary among different types of malignancies and therapy combinations [70,71]. They are mostly mild to moderate and they can be administered with immunosuppressive therapies [62,72,73]. However, studies have found that irAEs’ toxicity is associated with ICPis’ efficacy; thus, severe irAEs outbreak leads to an intensified and sustained anti-tumor immune response [74], whereas immunomodulatory agents and irAEs’ regression may compromise ICPi’s clinical benefit [75].

Understanding the multiple aspects underlying immunopathogenesis of irAEs can improve the management of these autoimmune dysregulations and support finetuning of ICPis’ treatments in cancers therapies. According to clinical reports, irAEs can affect almost any organ system, the most common ones involve the skin, endocrine glands, gastrointestinal system, lung, liver, and musculoskeletal systems, while less frequently are renal, cardiovascular, ocular, and neurological events [52,66,76]. However, there is a distinct organ-specific toxicity among these two types of monoclonal antibodies. It appears that colitis and hypophysitis are more common irAEs when treated with anti-CTLA-4, whereas pneumonitis and thyroiditis occur more frequently with anti-PD-1 therapy [72]. Moreover, although irAEs typically have a delayed onset and prolonged duration when compared with chemotherapy toxicity, there are differences in delay and duration among ICPis therapies. For example, colitis appears faster following anti-CTLA-4 and slower after anti-PD-1 therapy [72,76]. Genetic predisposition can explain the diversity and susceptibility to distinct ADs, for instant specific haplotypes of human leukocyte antigen (HLA) have been associated with certain ADs. In fact, patients with preexisting autoimmune disorders, disturbing immune tolerance with ICPis treatment, can significantly increase the potential risk factors to disease exacerbations [77,78]. Furthermore, variation in gastrointestinal flora was found to play a role in host immunity [79], and certain alterations in gut microbiome composition of metastatic melanoma patients who went through ipilimumab therapy were found to increase the risk to develop ICPi’s induced colitis [80,81]. Moreover, patients with ipilimumab-induced colitis had elevated levels of IL-17 [72], substantial increase in IFN-γ and TNF-α [49], a broad expansion of CD4 Th17 cells and diminished activity of regulatory T cells (Tregs).

Studies in patients with irAEs have shown that anti-CTLA-4 therapy mediates a greater diversification in T-cell repertoire, resulting in autoreactive T cells [82], whereas anti-PD-1/anti-PD-L1 treatments modulate humoral immunity as well, and thereby enhance pre-existing and new auto-antibodies [72,78].

Immunogenic pathways of self-tolerance breakdown, mediated by autoreactive T cells, can be through cross-reactive tumoral antigenicity, where the targeted T-cell antigens are found in both tumor and normal tissue [69]. Alternatively, epitope spreading can come about upon tumor cells’ pyroptosis. Destruction of cancer cells releases multiple antigens that are processed and presented by APCs, expanding the T cell repertoire. Epitope spreading occurs when some of the newly recruited T cells are against self-antigens, some of which are cryptic antigens that are not present during T cells’ negative selection in the thymus. Extensive activation of additional T cells and their clonal diversification can activate non-specific T cells, which in some cases, target self-antigens and lead to bystander activation [71].

### 1.6. The Place of PD-1, CTLA-4 in CD

Facing a processed peptide presentation, the coinhibitory ‘checkpoints’, PD1 [83] and CTLA-4 [84], are recruited to regulate T cells. A deficiency or disfunction in either of the two is involved in AD development [2] and is suggested to drive CD. In fact, CTLA-4 polymorphisms confer susceptibility to CD [85]. Being a substantial actor in the adaptive response, soluble CTLA-4 is increased in CD patients with associated ADs [86] and correlates with intestinal damage [87]. It can be summarized that a functional CTLA-4 is a protective gene in CD.

Similarly, the PD-1/PDL1 dysregulated pathway was shown to contribute to the pathogenesis of CD [88]. Interestingly, alternative PD1 splicing was shown in CD patients, suggesting a complex pathophysiology that occurs in CD patients [89]. The authors hypothesized that those different isoforms of PD1 could contribute to CD development.

Understanding the mechanisms of the two costimulatory molecules’ functional integrity, in alleviating the risk for CD, or on the contrary, their dysfunction in CD genetic susceptibility might help unravel the role played by the ICPi in CD induction. It appears that cancer patients treated by Nivolumab (an anti-PD-1) and by Ipilimumab (an anti-CTLA-4) were described to develop CD.

## 2. Nivolumab (Anti PD-1) + Ipilimumab (Anti-CTLA) Induced CD

Reviewing the literature, 13 patients were reported with ICPi- associated CD. Table 1 summarizes the literature on the reported cases of ICPi associated CD.

Analyzing the table, crucial information is missing for reliable CD diagnosis [8,97]. A critical assessment shows that several diagnostic criteria were not fulfilled. Not all the described patients were IgA-TTG positive [10,98], and AGA positivity is not specific for CD [99]. The number of IEL [100] is not mentioned for all the patients; not all patients had an intestinal histological evaluation; for none of the patients, the most reliable genetic marker, HLA-DQ2/8, was determined; gluten withdrawal was only partially helpful in some of the patients.

A primary histological diagnostic concern is the similarity between ICPi-induced CD and the frequently described CPi-induced duodenitis. The duodenal immune-related adverse events might resemble or even be indistinguishable from a plethora of other gastrointestinal conditions, such as viral or bacterial small bowel infections, Crohn’s duodenitis, autoimmune enteropathy, CD, drug-induced duodenal injury, and some other entities [101,102,103,104,105]. This is the logic behind the scientific concern and the reason for the current call for a more informative description when this drug-induced CD is suspected.

## 3. Potential Mechanisms for ICPi Celiac Disease Induction

### 3.1. Failure of the Immune Checkpoints to Execute Their Moderative and Tolerogenic Functions in Predisposed/Potential CD

As mentioned above, the ICPis play a major role in regulating the immune homeostasis and suppressing self-reactive T cells, thus maintaining central and peripheral tolerance and counteracting over-reaction and hyperstimulation of the reactive and the innate immune cells [2,83,84,106]. CD is not an exception [88,89,107,108,109,110,111]. When suppressed by their specific inhibitors, the ICPis cannot fulfill their role, thus opening the window for CD induction [88,89,90,91,92,93,94,95].

### 3.2. Preexisting CD-Associated ADs Flare-Up after ICPi Delivery

As part of the mosaic of autoimmunity [112], CD is associated with multiple ADs and many genes and autoantibodies are shared between them. It appears that many of those ADs (RA, psoriasis, IBD, Lupus, Graves’ diseases, Hashimoto disease, vitiligo, autoimmune hepatitis, multiple sclerosis) flare-up upon ICPi therapy [113]. Once again, alluding to those drugs’ capacity for autoimmunity induction [109,114].

### 3.3. ICPi Enhances B-Cell Clonality and Autoantibody Secretion

Blockade of PD-1/PD-L1 pathways increased B-cell proliferation and immunoglobulins secretion [115,116]. Interestingly, ICPi therapy induces new autoantibodies, such as anti-thioperoxides and anti-thyroglobulin [117]. As a proof of concept, 12/13 patients (Table 1) developed anti-tTG, a well-established autoantibody of CD [10,98]. The relations between ICPi administration, gliadin-induced CD4 T-cell selective clonality and tTG-producing B-cell expansion, remain to be established to fortify the cause and effects relationship.

### 3.4. Cross-Presentation of Shared Antigens

In this phenomenon, tumor originated neoantigens or dead bystander cells released antigens could be taken up by APCs, processed and presented to T cells, thereby activating an additional wave of T cells that can attack normal tissue. In the present case, those neo-antigens-induced T-cells can target the small bowel and induce CD-like pathology and potentially increased tTG auto-antibodies [116].

### 3.5. Spreading of the Epitope to the Small Bowel Tissue

Epitope spreading is defined by the development of an immune response to epitopes distinct from, and non-cross-reactive with, the disease-causing epitope [118]. In our case, epitopic diversification can spread from the costimulatory molecules to an enteric epitope. Indeed, epitope spreading was reported after CTLA-4 blockage [119].

### 3.6. Genetic Predisposition

Autoimmune diseases have always a genetic predisposition and are triggered by environmental factors, including drugs [120]. It is foreseeable that cancer patients who are genetically predisposed to autoimmunity may be more prone to develop CD upon ICPi usage [116]. Unfortunately, no CD-specific genetic screening was performed in the patients presented in Table 1.

### 3.7. The Microbiome Composition and Diversity

Autoimmune diseases are heavily microbiome and metabolome dependent [121] and in CD patients, gluten, but also gluten-free diet, are known to affect the microbiome/dysbiome ratio [122]. Furthermore, the therapeutic responsiveness to ICPi, as well as irAEs severity, are closely associated [116,123], and commensal bacteria facilitate anti-PD-L1 efficacy [124]. Interestingly, specific microbiome is beneficial while dysbiome is detrimental for the ICPi outcome and adverse effects development [125].

### 3.8. Local Intestinal Expression of the Immune Checkpoints

The gastrointestinal tract is the most abandonly inhabited immune system, where extensive APCs-T cells interactions and cross-talks are taken place. It can be assumed that the mucosal compartment contains a heavy load of checkpoints/costimulatory molecules. This concentrated organ-specific expression might contribute to the development of the enteric-specific irAEs [116].

### 3.9. Direct IPCi Toxic Side Effects

ICPis-induced duodenitis is a well-described histological adverse effect [101,102,103,104,105] that could be confused with CD enteric pathology. Intriguingly, those severely ill cancer patients have pains and consume non-steroidal anti-inflammatory and anti-pain drugs, known to induce duodenitis [126].

### 3.10. Pathological Duodenal Manifestation of the Primary Oncological Diseases, Chemo- and Immunotherapy

The cancer processes may invade the duodenum; chemotherapy, radiotherapy, immunosuppressive, and immunotherapy can affect the duodenal morphology and function, thus, increasing perplexity of the upper intestinal histological interpretation [127,128].

### 3.11. Potential Cross-Reactive Antibodies between the ICPi and Intestinal Antigens

The potential presence of cross-reactive antibodies against an external/environmental factor that shares epitopes with an intestinal/duodenal antigen might be a powerful mechanism for molecular mimicry that drives autoimmunity [129,130,131]. In the current case, the antibodies created against ICPi might, for example, cross react with the PD-1, CTLA-4, tTG, or tight junction proteins, thus driving CD.

### 3.12. Autoimmunity Due to Loss of Treg Homeostasis

Regulatory T-cells (Tregs) are essential for immune balance and can play a dual function facing cancer evolvement. Their “Yin–Yang” interplay between modulation of autoimmunity and fighting cancer immunity will affect the patient’s outcome. The Tregs number was shown to negatively correlate with ICPis’ induced IRAEs [34]. Despite the central role played by Tregs in CD pathophysiology [132], this phenomenon was never checked in ICPi-induced CD.

## 4. Discussion

Many ADs were described to flare up under ICPi anti-cancer treatment. The ICPi induced IRAEs are primarily T cell mediated, may affect various organs, are expressed in different times along the immune therapy, hence, have other primary events or mechanisms [116]. It is assumed that early onset immune adverse effects are more likely to be an autoinflammatory reaction caused by systemic immune disbalance. However, the late-onset ones are more of an autoimmune type of reaction [116]. Undoubtedly, gut-targeted broken organ-specific self-tolerance is the primary mechanism driving the presently described CD autoimmunity.

Although CD is a classical autoimmune condition, it was sporadically reported as an IRAE, mainly in case reports. The present review summarizes the 13 cases described so far, where several characteristic features can be identified. There is a delayed onset between the immunotherapy and the CD symptomatic appearance. The clinical presentation is different from the known side effects of chemotherapy. It is not dependent on the type of the ICPi treatment. A high susception index, early detection and appropriate intervention with gluten withdrawal and immune suppression are essential for the patient’s disease outcome [76,116].

For a definitive diagnosis, CD-associated serology should be above the upper normal limits and duodenal biopsies should show the classical diagnostic histological abnormalities [10,100]. Since ICPi-induced duodenitis extensively overlaps the CD-associated duodenitis, it is recommended to take extra precautions, check additional CD-specific serologies and perform the HLA-DQ2\8, as genetic markers for CD. When diagnosed appropriately, the patients will respond to a gluten-free diet, but sometimes corticosteroid and/or immune-suppressive drugs should be added in refractory situations [133].

Specific autoantibodies are keys markers in the autoimmune definition. Studies of longitudinal profile assessment of autoantibodies in patients developing autoimmune ICPi-toxicity are lacking. Whether a new autoimmune mechanism underlines the ICPi-related autoimmunity, distinguishing it from the classical autoimmune condition is still an enigma. The report of early changes in B-cells in patients with increased risk of irAEs represents a preliminary step in this direction [78,134]. Elevated cytokines were suggested to predict ICPi immune toxicity. Increased IL-17 on Ipilimumab and gamma interferon-inducible CXCL-9 and CXCL-10, while on anti-PD1/PDL1 therapy, was shown to be associated with irAEs development [78,135,136]. No such studies were performed on the currently described ICPi-induced CD patients. Genetic background, epigenetics, additional environmental circumstances, microbiome/dysbiome ratio, luminal metabolomic repertoire, cytokine profile and enteric immune activation dynamics are still waiting to be explored.

Despite the fact that CD is a totally different gastrointestinal condition, inflammatory bowel disease was most recently described to flare up upon ICPi therapy [137,138,139]. The issue of a de novo or preexistent AD that flares up is still out of the jury.

## 5. Conclusions

Using ICPi to combat complicated or resistant cancer represents a huge step toward a better outcome, but the ensuing irAEs represent an alarming signal for the treating teams. Autoimmune complications are continuously described and CD took its place on the list. Figure 2 presents, schematically, some of those potential mechanisms and pathways that might operate in ICPi induced CD. Future work on the adverse ICPi effects should focus on identifying those at higher risk for developing Ads in general and CD in particular. Upon identification, appropriate management should be instituted. Gluten-free diet for long-term and immune-suppression only for as long as needed. Understanding the predisposing factors and exploring the mechanism and cellular pathways may lead to new therapeutic strategies expanding the arsenal of ICPi targets.

## Figures and Tables

**Figure 1 biomedicines-10-00609-f001:**
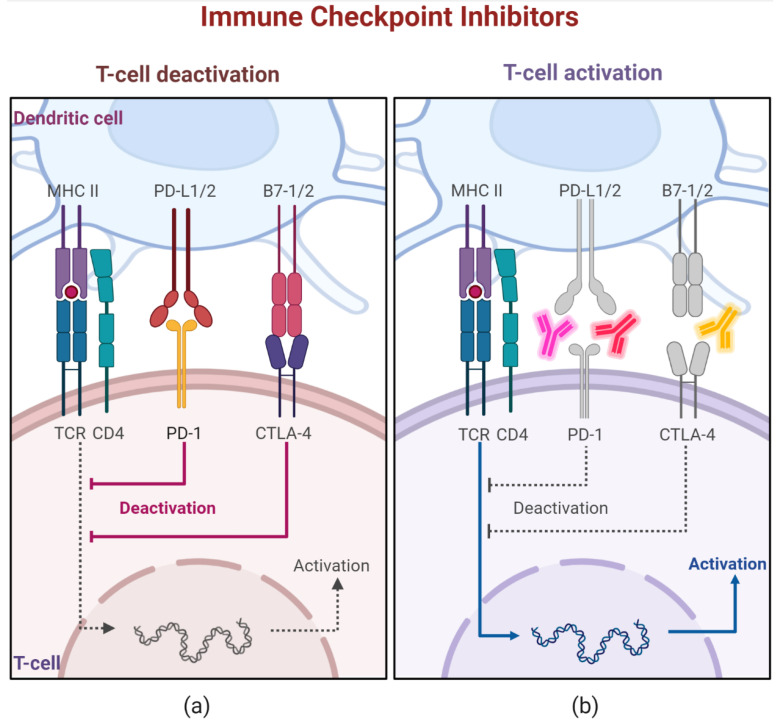
T cell activation through checkpoint inhibitors. A schematic presentation of anti-PD-1/PD-L1 and anti-CTLA-4 agents in action. (**a**) APCs, such as dendritic cells, present processed peptides to T cells on MHC molecules. Upon activation, T cell gradually expresses on its membrane the CTLA-4. When it binds to B7-1/2, it initiates co-inhibition pathways that lead to T cell anergy. In peripheral tissues, activated T cell can be de-activated by the binding of PD-1 to PD-L1 or PD-L2. (**b**) The anti-CTLA-4 and anti-PD-1/PD-L1 monoclonal antibodies block those inhibitory pathways resulting in effective anti-tumor T lymphocyte responses.

**Figure 2 biomedicines-10-00609-f002:**
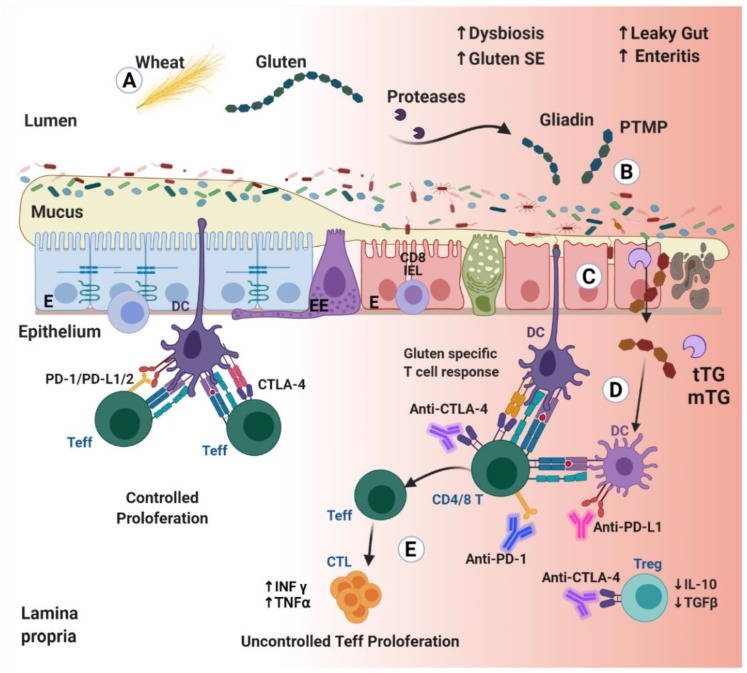
Immune checkpoint inhibitors and celiac disease. (**A**) Gluten is ingested and digested, reaching the gut lumen as gliadin peptides. (**B**) Gliadins are rich in glutamine and proline, thus are a prime substrate for deamidation and cross-linking by luminal and mucosal transglutaminases, thus, turning those naïve molecules into immunogenic ones. Transglutaminase capacity to deamidate or transamidate, results in increased post-translation modified proteins (PTMP). Luminal digestive peptidases cannot further break down those bonds, hence, inducing gut inflammation, mucus disruption and intestinal epithelial damage. (**C**) Gluten increases intestinal permeability by binding to epithelial CXCR3 receptors, resulting in zonulin release. Gliadin-transglutaminase transformed peptides can potentially infiltrate through the open junctions or trans-enterocytically into the lamina propria. A breach in the epithelial barrier exposes the highly immunoreactive sub-epithelium to luminal foreign antigens, stimulating the local immune system. (**D**) In the lamina propria, gliadin-transglutaminase cross-linked complexes induce pro-inflammatory cytokines. Two types of DC are present, sub-epithelial DCs that send protrusions into the lumen and sense the gut microbiota, and the lamina propria DCs that migrate to lymph nodes, where they present antigens and activate T cells. Immune checkpoint inhibitors block co-inhibitory pathways unleashing effector T cells and depleting regulatory T cells. (**E**) Uncontrolled activation and proliferation of cytotoxic T lymphocytes (CTLs) further aggravate barrier perturbation, secreting IFNγ and TNFα cytokines, leading to severe intestinal damage.

**Table 1 biomedicines-10-00609-t001:** The reported cases of ICPi associated CD.

ICPi	Gender/ Age (y)	CD Intestinal Pathology	CD Associated Serology			References
			tTG	AGA	EMA	
Nivolumab (anti-PD-1)	M/70	Crypt/villous ↑, IEL↑, inflammation, ulceration	+		+	[90]
Nivolumab+ Ipilimumab	W/74	Villous atrophy, IEL↑, inflammation	+			[91]
Ipilimumab (anti-CTLA-4)	M/62	IEL↑, crypt distortion	+	+		[92]
Pembrolizumab (anti-PD-1)	W/63	Crypt/villous ↑	-	+		[93]
	M/79	Villous atrophy, IEL↑, inflammation	+	+		[94]
Eight cases: Five anti PD1 One anti CTLA-4 Two combined	2 W, 6 M /44–73	6/8 were biopsied: villous atrophy, IEL↑, inflammation	+			[95]
No data are available.			?			[96]

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
