# Peer review of "Checkpoint Inhibitors and Induction of Celiac Disease-like Condition"

_biomedicines, 2022, doi:10.3390/biomedicines10030609_

Round 1
Reviewer 1 Report
Very interesting topic as side effects of moderd drugs are of great concern. Celiac disease triggered by new medications is not well known possibility and therefore spreading this knowledge is very needed.
Author Response
Hello
I was asked to reduce my references to a maximum of ten self-references
The reference number was reduced by 23 publications, from a total of 162 to 139 references
Best regards
Prof. Aaron Lerner , corresponding author

Reviewer 2 Report
This manuscript describes the background of the mergence of autoimmune disease induced by the use of immune check point inhibitors. This is very well elaborated and yields information even for a reader who is not so familiar with this new type of tumour therapy.
My only concern is the title of the manuscript. In the 13 patients who are the background of this manuscript, there is indeed a disease with villous atrophy and intraepithelial lymphocyte infiltration. However, this condition cannot be regarded as celiac disease, as the authors mention themselves. Important criteria for celiac disease are not fulfilled.
Therefore, I suggest to change the title to: Checkpoint inhibitors and induction of a celiac disease - like condition
Author Response
Hello, thanks for your valuable comment.
The title was changed
English and style were improved
Reviewer 3 Report
Authors review an interesting topic, as it is the adverse effects of immune checkpoint inhibitors in oncological therapy resistant cancer. The issue is receiving increasing attention in the literature, so readers will very much appreciate it. The structure of the review is very well structured and organized. In my opinion, the manuscript is suitable for publication.
Author Response
Hello, thanks for your comments
English and style were improved